# J Subgroup Avian Leukosis Virus Strain Promotes Cell Proliferation by Negatively Regulating 14-3-3σ Expressions in Chicken Fibroblast Cells

**DOI:** 10.3390/v15020404

**Published:** 2023-01-31

**Authors:** Moyu Wang, Hongmei Li, Xiyu Sun, Jianhua Qiu, Changhua Jing, Huiyue Jia, Yujie Guo, Huijun Guo

**Affiliations:** Shandong Provincial Key Laboratory of Animal Biotechnology and Disease Control and Prevention, College of Animal Science and Veterinary Medicine, Shandong Agricultural University, Tai’an 271018, China

**Keywords:** chicken 14-3-3σ, ALV-J, cell proliferation, cell cycle, G2/M-phase block

## Abstract

This study focuses on clarifying the regulation of chicken 14-3-3σ protein on the fibrous histiocyte proliferation caused by ALV-J-SD1005 strain infection. DF-1 cells were inoculated with 10^2^ TCID_50_ of ALV-J-SD1005 strain; the cell proliferation viability was dramatically increased and 14-3-3σ expressions were dramatically decreased within 48 h after inoculation. Chicken *14-3-3σ* over-expression could significantly decrease the cell proliferation and the ratio of S-phase cells, but increase the ratio of G2/M-phase cells in ALV-J-infected DF-1 cells; by contrast, chicken *14-3-3σ* knockdown expression could cause the opposite effects. Additionally, chicken *14-3-3σ* over-expression could also dramatically down-regulate the expressions of CDK2/CDC2, but up-regulate p53 expressions in the DF-1 cells; in contrast, the knockdown expression could significantly increase the expressions of CDK2/CDC2 and decrease p53 expressions. It can be concluded that chicken 14-3-3σ can inhibit cell proliferation and cell cycle by regulating CDK2/CDC2/p53 expressions in ALV-J-infected DF1 cells. ALV-J-SD1005 strain can promote cell proliferation by reducing *14-3-3σ* expressions. This study helps to clarify the forming mechanism of acute fibrosarcoma induced by ALV-J infection.

## 1. Introduction

Avian leukosis virus (ALV) is a tumorigenic retrovirus and is divided into 11 subgroups, whose names range from A to K [1,2]. Among these subgroups, J subgroup of ALV (ALV-J) was first reported in 1991 and causes huge economic losses to the world poultry industry every year due to its high variability and strong pathogenicity [3]. ALV-J was early isolated from broiler breeding chickens and caused myelocytoma [4,5]. It was subsequently detected in egg-laying chickens, wild birds, etc., and caused multi-tissue hyperplasia [6]. In 2012, an ALV-J strain containing *v-fps* oncogene (ALV-J-SD1005) was isolated from the local broiler breeding chickens named “817” in China. It can cause rapidly fibrous tissue hyperplasia within 2~3 weeks and 100% mortality in 3~4 weeks in the infected chickens, which brings a great challenge for control ALVs [7].

In recent years, many studies have indicated that some host molecules such as vascular endothelial growth factor (VEGF), doublecortin-like kinase 1 (DCLK1), phosphoinositide dependent protein kinase 1 (PDPK1), cyclindependent kinase inhibitor 1B (CDKN1B), Interleukin-6 (IL-6), micro-RNAs, etc., participate in the cell proliferation or tumorigenesis induced by some ALV-J strains [8,9,10,11,12]. However, little is known about the mechanism of acute fibrous tissue hyperplasia or fibrosarcoma induced by the ALV-J strains such as ALV-J-SD1005. In 2022, Liu et al. reported that 14-3-3σ, as a cell cycle regulator, was dramatically down-regulated in chickens with subcutaneous fibrous tissue hyperplasia caused by ALV-J-SD1005 strain, indicating that it might participate in the formation and development of acute fibrosarcoma [13].

14-3-3 σ, as a member of the highly conservative 14-3-3 protein family, mainly exists in epithelial cells and can promote G2-phase block of the cell cycle after DNA damage by interacting with a variety of signal proteins such as CDK2, CDC2, etc., and regulate cell growth, differentiation and proliferation in mammal cells [14,15,16]. It also acts as a key regulator of p53 to control mitotic progression in response to DNA damage and inhibits the development of several substantial tumors, such as nasopharyngeal, colorectal and breast cancers [17,18,19,20,21]. In some studies on lung and gastric cancers, 14-3-3σ was found to show high expression in cancer tissues, promoting the progression of tumorigenesis [22,23,24]. However, there are no reports on the roles of 14-3-3σ in the fibrous tissue hyperplasia caused by ALV-J-SD1005 strain.

In this study, chicken 14-3-3σ expression and cell proliferation in the DF-1 cells inoculated with ALV-J-SD1005 strain were detected at different times and the roles of 14-3-3σ on the cell proliferation induced by ALV-J-SD1005 strain was analyzed by transfection experiments in vitro. It was found that ALV-J-SD1005 strain could promote cell proliferation and inhibit chicken 14-3-3σ expressions, and chicken 14-3-3σ could negatively regulate the cell cycle of ALV-J-infected cells and cause G2/M-phase block and down-regulate CDK2/CDC2 expressions, and up-regulate p53 expressions. This is a first report on the regulation effects of chicken 14-3-3σ in the fibrous histiocyte proliferation induced by the ALV-J-SD1005 strain.

## 2. Materials and Methods

### 2.1. Cells and Virus

DF-1 cells, a chicken fibrous cell line which can be used for exogenous ALV replication, were grown in DMEM medium (Bio-Channel, Nanjing, China) with 10% fetal bovine serum (GIBCO, Grand Island, NE, USA) at 37 °C and 5% CO_2_ concentration, kindly donated by Prof. Zhizhong Cui. ALV-J-SD1005 strain was donated by Prof. Shuhong Sun and maintained in this laboratory; the *v-fps* oncogene in its genome was confirmed by PCR [7]. The infectious dose of virus halves in the tissue culture was determined using the limited dilution method on 96-well plates covering DF-1 cells according to the Reed–Muench method.

### 2.2. Viral Infection

DF-1 cells were inoculated with 10^2^ TCID_50_ of ALV-J-SD1005 strain, and 2 h after inoculation (hpi) the viral solution was discarded and replaced with fresh maintenance medium containing 1% fetal bovine serum (FBS), while the cells without virus-inoculation were set up as controls. The samples of cells and cell supernatants were collected at 0, 12, 24, 36 and 48 hpi, respectively; the cells were used to extract total RNA and protein. qPCR and ELISA methods were used to detect the replication of virus in DF-1 cells and the titers of virus antigen in the cell supernatants, respectively; qPCR and Western blot methods were used to detect the expressions of 14-3-3σ mRNA and protein levels. 

This experiment and all the animal materials in this study were approved by the Animal Ethics Committee of Shandong Agricultural University Animal Protection and Welfare Institute (Number: SDAUA-2021-054). All procedures related to the animals and their care conformed to the internationally accepted principles found in the Guidelines for Keeping Experimental Animals issued by the government.

### 2.3. Reverse Transcription and qPCR

Total cellular RNA was extracted using a kit from Bori (Hangzhou, China). RNA concentration was determined using a spectrophotometer and 1 μg RNA was used as a template and reverse transcribed to cDNA using a kit from Tiangen (Beijing, China) according to the manufacturer’s instructions. All qPCR assays were performed using the ABI 7500 sequence detection system (Applied Biosystems) instrument, which used SYBR Green reagent (YEASEN, Shanghai, China) for cDNA amplification with a total reaction system of 10 μL and specific primers designed by Primer Premier 6.0 (Premier, Canada) based on NCBI sequences as shown in Table 1, with GAPDH as the internal reference gene. The reactions were performed using the following procedure: pre-denaturation 95 °C for 10 min; annealing/extension 95 °C for 15 s and 60 °C for 60 s for 40 cycles; and addition of melting curves 95 °C for 15 s, 60 °C for 60 s and 95 °C for 15 s. To quantify the PCR products, the relative expression level of each gene was calculated and normalized using the 2^−ΔΔCT^ algorithm [25].

### 2.4. Western Blot Analysis

To extract total protein from treated DF-1 cells using RIPA lysis buffer (Beyotime, Shanghai, China), protein concentration was determined using the BCA protein assay kit (Beyotime, Shanghai, China), and 20 μg of protein was separated on 12% SDS polyacrylamide gels and transferred to 0.45 μm polyvinylidene difluoride (PVDF) membranes (Biosharp, Anhui, China). The membranes were closed in 5% skim milk powder (Solarbio, Beijing, China) for 2 h at 37°C and then incubated overnight at 4 °C with diluted primary antibodies including anti-P53 (AF7671, Beyotime, Shanghai, China), anti-CDC2 (AF0111, Beyotime, Shanghai, China), anti-CDK2 (AF1063, Beyotime, Shanghai, China), anti-β-actin (AT0001, Engibody, Milwaukee, WI, USA) and laboratory-made rabbit anti-14-3-3σ polyclonal antibody. Then, TBST was washed at room temperature for 10 min and incubated with horseradish peroxidase-conjugated secondary antibody (CW0102, CWBIO, Beijing, China) at 37 °C for 1 h. Finally, protein bands were visualized using an ultrasensitive ECL chemiluminescence kit (NCM Biotech, Suzhou, China).

### 2.5. ELISA for Viral Antigen Titers

To detect the titer of virus in cell supernatant samples, ALV antigen was detected using an ALV P27 antigen ELISA kit (IDEXX, Beijing, China) according to the manufacturer’s instructions as follows: 100 μL of the sample to be tested was added to each well, and negative and positive controls were set up and incubated for 1 h at 37 °C, followed by 4 washes with washing solution, after which 100 μL of enzyme-labeled antibody was added to each well and incubated for 1 h at 37 °C, followed by 4 washes with washing solution. S/P value = (sample OD_450nm_ value − negative OD_450nm_ value)/(positive OD_450nm_ value − negative OD_450nm_ value).

### 2.6. Amplification of Chicken 14-3-3σ Gene

The primers (forward: 5′-CGCGGATCCAGCCCCACTCCTCGCCCCGAC -3′; reverse: 5′-CCCAAGCTTTCAGTTCTTGGGCTCTTCGCC-3′) were designed and synthesized to amplify the whole 14-3-3σ open reading frame (ORF) gene according to the chicken 14-3-3σ sequence (NM_001293176.1) in GenBank. The chicken liver total RNA was extracted using an RNA extraction kit and reverse transcribed to obtain cDNA encoding chicken 14-3-3σ. The target genes were amplified from the obtained cDNA by polymerase chain reaction (PCR) under the following conditions: pre-denaturation at 95 °C for 5 min; denaturation at 95 °C for 30 s, annealing at 55 °C for 60 s, extension at 72 °C for 50 s, 30 cycles; and then 72 °C for 10 min. The PCR products were analyzed using 1% agarose gel electrophoresis and sequenced by Shanghai Bioengineering Technology Service Co.

### 2.7. Over-Expression and Knockdown Expression of Chicken 14-3-3σ

The amplified chicken 14-3-3σ gene was cloned into the pcDNA3.1 vector and identification of the recombinant plasmid was achieved using enzymatic digestion and sequencing. A small interfering RNA (siRNA) against chicken 14-3-3σ (14-3-3σ siRNA) with the target sequence 161, 5′-CCCUCCAGGCCGAGCGCUGGC-3′ was designed and synthesized (Biotech, Shanghai, China). After the DF-1 cells in the 6-well plate grew up to 60%, pcDNA3.1-14-3-3σ plasmids, 14-3-3σ siRNA and negative control siRNA (5′-UUCUCCGAACGUGUCACGUTT-3′) were transfected using Lipofectamine 3000 transfection reagent (Invitrogen, Carlsbad, CA, USA) according to the manufacturer’s instructions, respectively. The cells were collected at 24 h post-transfection (hpt); total RNA and protein were extracted and the over-expression and knockdown efficiencies were determined by Western blot and qPCR.

### 2.8. Cell Counting Kit 8 Assay

DF-1 cells were inoculated with 10^2^ TCID_50_ of ALV-J-SD1005 strain and the cells without virus inoculation were set up as controls. The cell proliferation viability was detected using CCK-8 kits (Biosharp, Anhui, China) at 12, 24, 36 and 48 hpi. The pcDNA3.1-14-3-3σ plasmids, 14-3-3σ siRNA plasmids and negative control plasmids were transfected into virus-infected DF-1 cells at 12 h, respectively; the cell proliferation viability of ALV-J-SD1005-infected cells was examined using CCK8 kits at 12, 24, 36 and 48 hpt. 10 µL of CCK-8 reagent was added into each well and the absorbance was measured at OD_450nm_ using a microplate reader (Bio-Rad) after incubation at 37 °C for 2 h. The cell viability calculation formula was as follows: (experimental group − blank group)/(control group − blank group) × 100%. All experiments were repeated at least six times.

### 2.9. EDU Assay

The cell samples from the above treatment at 36 hpi or hpt were incubated at 25 °C for 2.5 h using 5-ethynyl-2′-deoxyuridine (EDU) reagent (Beyotime, Shanghai, China) according to the manufacturer’s instructions and then fixed in paraformaldehyde at 25 °C for 15 min, rinsed with PBS 3 times and incubated with 50 μL of the configured click reaction solution at 25 °C for 30 min and protected from light; finally, the nuclei were stained with DAPI and the cells were analyzed under fluorescence microscopy. The images and numbers of EDU-stained cells were analyzed under fluorescence microscope. All experiments were repeated three times.

### 2.10. Cell Cycle Analysis

The virus-infected DF-1 cells in 6-well plates were transfected with pcDNA3.1-14-3-3σ plasmid, pcDNA3.1-His plasmid, 14-3-3σ siRNA and NC siRNA, respectively; the cell samples were collected at 36 hpt and washed twice with pre-chilled PBS, fixed with 70% ethanol overnight at 4 °C, subjected to cell cycle assay using a cell cycle assay kit (Beyotime, Shanghai, China) for PI staining and incubated at 37 °C for 30 min protected from light. The cells were examined using a flow cytometer (BD Biosciences, Franklin Lakes, NJ, USA) and the data were analyzed with flowjo software (v10, Leonard Herzenberg, Ashland, OR, USA). All experiments were repeated three times.

### 2.11. Statistical Analysis

All the data are expressed as the mean ± standard deviation. The significant differences between groups were determined using Student–Newman–Keuls tests of multiple comparisons. *p* < 0.05 was considered statistically significant. *p* < 0.01 was considered highly significant. 

## 3. Results

### 3.1. ALV-J-SD1005 Strain Can Rapidly Replicate and Release in DF-1 Cells

To analyze the effects of ALV-J infection on chicken fibroblast cell proliferation and the expressions of chicken 14-3-3σ, DF-1 cell (a chicken fibroblast cell line) was inoculated with ALV-J-SD1005 strain as an infection cell model. The viral proliferation viability in DF-1 cells was firstly assessed at different times post-inoculation. The results in Figure 1A show that ALV-J gp85 mRNA levels in ALV-J-infected DF-1 cells were significantly higher than those in the control group at 12, 24, 36 and 48 hpi, and gradually increased with the increase in infection time; meanwhile, ALV P27 antigen titers in the cells supernatant were also significantly higher than those in the control group up to the higher values (0.76 ± 0.08) at 48 hpi (Figure 1B). These indicate that ALV-J-SD1005 virus can rapidly proliferate and release in DF-1 cells.

### 3.2. ALV-J-SD1005 Strain Inhibits Chicken 14-3-3σ Expressions in DF-1 Cells

14-3-3σ expressions in the DF-1 cells infected with ALV-J-SD1005 strain were detected using qPCR and Western blot methods at different times post-inoculation. The results show that 14-3-3σ mRNA expressions in ALV-J-infected cells at 24 and 36 hpi were significantly decreased (Figure 2A), and the protein expressions were also significantly decreased from 24 hpi to 48 hpi compared with those in the control groups (Figure 2B). These results indicate that ALV-J-SD1005 strain can dramatically down-regulate 14-3-3σ expressions in DF-1 cells.

### 3.3. ALV-J-SD1005 Strain Promotes the Division and Proliferation of DF-1 Cells

The proliferation viability of DF-1 cells was detected using CCK-8 assay at different times post-virus inoculation. The results in Figure 3A show that the cell proliferation viabilities in ALV-J-SD1005-infected groups at 12, 24 and 36 hpi were significantly higher than those in the control groups. The results of EDU fluorescence staining in Figure 3B also show that the EDU-positive cells in ALV-J-SD1005-infected group at 36 hpi were clearly more than those in the control group. These indicate that ALV-J-SD1005 virus can promote the cell proliferation of DF-1 cells.

### 3.4. Construction and Bioactivity Analysis of Chicken 14-3-3σ Over-Expression Plasmid

To analyze the roles of chicken 14-3-3σ protein on the cell proliferation induced by ALV-J-SD1005 infection, the specific primers were designed and synthesized to amplify 14-3-3σ gene with the size of 792 bp from chicken liver (Figure 4A); a recombinant eukaryotic expression plasmid containing chicken 14-3-3σ gene (pcDNA3.1-14-3-3σ) was constructed using the pcDNA3.1 plasmid containing His-tag (Figure 4B) and identified by double digestion test and nucleic acid electrophoresis analysis. The results show that an enzymatic fragment of the same size as the target gene was obtained (Figure 4C lane 2). The pcDNA3.1-14-3-3σ plasmid and the control plasmid (pcDNA3.1-His) were transfected into DF-1 cells, respectively; after 24 hpt, total RNA and total protein in the cells were extracted to detect the over-expression efficiency by qPCR and Western blot methods. The results showed that both mRNA and protein levels of chicken 14-3-3σ were significantly increased in pcDNA3.1-14-3-3σ groups (Figure 4D,E), indicating that chicken 14-3-3σ over-expression plasmid was successfully prepared.

### 3.5. Preparation of the siRNA against Chicken 14-3-3σ

An interfering RNA (siRNA) against the 161^st^ base site of chicken 14-3-3σ gene (14-3-3σ siRNA) and a negative control siRNA (NC siRNA) were designed and synthesized according to the design principle of siRNA. They were transfected into the DF-1 cells, respectively. After 24 hpt, total RNA and total protein were extracted to detect the knockdown efficiency. The results show that both mRNA and protein expressions of chicken 14-3-3σ were significantly reduced and the knockdown efficiency was up to 65~70% (Figure 5A,B), indicating that the prepared small interfering RNA can efficiently knockdown the expression of chicken 14-3-3σ.

### 3.6. Over-Expression of Chicken 14-3-3σ Inhibits the Cell Proliferation in ALV-J-Infected DF-1 Cells, but the Knockdown Expression Promotes That

To determine the roles of chicken 14-3-3σ on the cell proliferation caused by ALV-J infection, DF-1 cells were inoculated with 10^2^ TCID_50_ of ALV-J-SD1005 strain; after 12 hpi, pcDNA3.1-14-3-3σ plasmid and 14-3-3σ siRNA were transfected into the DF-1 cells, respectively. The cell proliferation viability was detected at 12, 24, 36 and 48 hpt. The results show that the cell proliferation viability in pcDNA3.1-14-3-3σ group from 24 hpt to 48 hpt was significantly lower than that in the control groups (Figure 6AI); the results of EDU fluorescence assay at 36 hpt also show that the ratio of EDU-positive cells in pcDNA3.1-14-3-3σ group was less than that in the control group (Figure 6AII-1/2), indicating that over-expression of chicken 14-3-3σ can inhibit the proliferation of ALV-J-infected DF-1 cells.

In Figure 6B, the results show that the cell proliferation viability in 14-3-3σ siRNA group from 24 hpt to 48 hpt was significantly higher than that in the control groups (Figure 6BI); additionally, the ratio of EDU-positive cells in 14-3-3σ siRNA group at 36 hpt was more than that in the control group (Figure 6BII-1/2). These results indicate that knockdown expression of chicken 14-3-3σ can promote the proliferation of ALV-J-infected DF-1 cells.

### 3.7. Chicken 14-3-3σ Over-Expression Causes G2/M-Phase Block of ALV-J-Infected Cells, Whereas Its Knockdown Expression Relieves That

To analyze the effects of chicken 14-3-3σ protein on the cell cycle, DF-1 cells of different division stages at 36 hpt were examined using flow cytometry. The results in Figure 7 show that the ratio of G2/M-phase cells in pcDNA3.1-14-3-3σ group was significantly higher than that in the control group, while the ratio of S-phase cells was significantly lower (Figure 7A); in contrast, the ratio of G2/M-phase cells in 14-3-3σ siRNA group was significantly lower than that in the control group and the ratio of S-phase cells was significantly higher (Figure 7B). These results suggest that chicken 14-3-3σ protein can inhibit DNA synthesis and cause G2/M-phase block in the division cycle of ALV-J-infected cells.

### 3.8. Chicken 14-3-3σ Over-Expression Significantly Down-Regulates CDK2/CDC2 Expressions and Up-Regulates p53 Expressions, While Its Knockdown Expression Causes the Opposite Effects

To clarify the regulation pathway of chicken 14-3-3σ on the cell cycle, the expressions of some signal proteins, including p53, CDC2 and CDK2, in the DF-1 cells transfected with different plasmids or siRNAs at 36 hpt were detected. The results of qPCR show that mRNA expressions of CDK2 and CDC2 in pcDNA3.1-14-3-3σ group were significantly decreased and p53 mRNA expression was significantly increased compared with those expressions in the control plasmid group (Figure 8AI). The results of Western blot show that protein expressions of CDK2 and CDC2 in pcDNA3.1-14-3-3σ group were significantly reduced and p53 protein expression was significantly improved (Figure 8AII-1/2). In contrast, in 14-3-3σ siRNA group, both mRNA expressions and protein expressions of CDK2 and CDC2 were significantly increased, while p53 expressions were significantly decreased (Figure 8BI,BII-1/2). These results suggest that chicken 14-3-3σ can regulate mRNA level and protein level expressions of CDK2, CDC2 and p*53* in ALV-J-infected DF-1 cells.

## 4. Discussions

ALV-J-SD1005 strain from the fibrosarcoma tissue of “817” broiler breeder chickens in Shandong belongs to subgroup J avian leukosis virus [26]. Unlike other ALV-J strains, it can induce the rapid occurrence and growth of fibrosarcoma in the infected chickens and cause high tumor incidence and mortality [13,27]. It was further confirmed that the tumorigenic gene *v-fps* had been implanted into its genome, which is presumed to be the main cause of the acute fibrosarcoma formation [7], but its tumorigenic process is still unclear. In our previous studies, it was found that ALV-J-SD1005 strain can not only cause the acute fibrosarcoma in the infected chickens within 2~3 weeks, but also cause the significant decrease of 14-3-3σ expressions in some tissues, such as fibrosarcoma and liver [13], suggesting that 14-3-3σ might be involved in the tumorigenic process.

In this study, DF-1 cells, as a chicken fibroblast cell line, were inoculated with ALV-J-SD1005 strain for assessing the effects of the virus on the cell proliferation. The results show that the virus can rapidly replicate and release in DF-1 cells, and the virus antigen titers in the supernatant can reach the higher values within 48 h; more noteworthy than this is the fact that the virus can dramatically promote the cell proliferation of DF-1 cells and significantly decrease chicken 14-3-3σ expression, which is consistent with the results in the infected chickens [13]. Therefore, DF-1 cells can be used as an ideal cell line for analyzing the roles of chicken 14-3-3σ on the cell proliferation induced by ALV-J-SD1005 strain.

Little has been known about chicken 14-3-3σ until now. To obtain its over-expression plasmid, chicken 14-3-3σ gene containing open reading frame was amplified from chicken liver and a recombinant eukaryotic expression plasmid (pcDNA3.1-14-3-3σ) was constructed. Then, a small interfering RNA used for knockdown expression of chicken 14-3-3σ was designed and synthesized. The bioactivities targeting chicken 14-3-3σ in DF-1 cells were confirmed by transfection tests.

In mammals, 14-3-3 σ, as a negative regulator, inhibits cell proliferation [28,29]. In this study, it was found that the proliferation viability of ALV-J-infected DF-1 cells was significantly decreased after over-expression of chicken 14-3-3σ; by contrast, it was significantly increased after its knockdown expression, suggesting chicken 14-3-3σ can negatively regulate the cell proliferation, similar to that reported in mammals. Combined with the results that chicken 14-3-3σ expressions were significantly decreased by ALV-J-SD1005 infection in DF-1 cells and in the infected chickens reported in our previous study [13], it is deduced that ALV-J-SD1005 strain can promote the proliferation of fibrosarcoma cells by reducing 14-3-3σ expressions.

Many studies show that, both in normal cells and in tumor cells, 14-3-3σ increase or over-expression can inhibit DNA-synthesis in the cell cycle and lead to cell division stuck in G2/M-phase, also known as G2/M-phase block; conversely, 14-3-3σ decrease or knockdown expression can relieve G2/M-phase block and promote cell division [30,31]. In ALV-J-infected DF-1 cells, chicken 14-3-3σ over-expression significantly decreases the proportion of S-phase cells and increases the proportion of G2/M-phase cells; by contrast, its knockdown expression significant increases the proportion of S-phase cells and decreases the proportion of G2/M-phase cells, suggesting that chicken 14-3-3σ can inhibit DNA-synthesis and cause G2/M-phase block in ALV-J-infected cells. Therefore, it is speculated that ALV-J-SD1005 strain promotes cell proliferation by relieving DNA-synthesis inhibition and G2/M-phase block caused by chicken 14-3-3σ.

It is well known that each stage of the cell cycle is regulated by some cyclins, among which CDK2 and CDC2 can promote the cell division from G1-phase to S-phase and from G2-phase to M-phase, respectively [21,29,30]. Many studies show that 14-3-3 σ can prevent CDK2 and CDC2 from entering the nucleus by combining or degrading them in the cytoplasm and inhibit DNA-synthesis and/or cause G2/M-phase block [29,30,32]. In this study, it was found that chicken 14-3-3σ over-expression significantly decreased CDK2 and CDC2 expressions and its knockdown expression significantly increased their expressions. Therefore, we deduce that it is by down-regulating the expressions of CDK2/CDC2 that chicken 14-3-3σ causes DNA-synthesis inhibition and G2/M-phase block; by contrast, ALV-J-SD1005 strain can relieve DNA-synthesis inhibition and G2/M-phase block to promote cell proliferation by regulating the cell cycle response mediated by chicken 14-3-3σ. 

In normal cells, p53 protein, as a negative regulator of the cell cycle, inhibits DNA-synthesis of S-phase in response to DNA damage [33,34]; additionally, it also participates in G2/M-phase block by targeting 14-3-3σ protein [30]. Meanwhile, 14-3-3σ can also improve the stability and transcriptional activity of p53 in a feedback [35]. However, many studies confirmed that p53 is abnormally increased as mutant p53 forms in some tumor tissues or cells [36], and does not possess a negative regulatory effect on cell proliferation [37,38]. Some studies also confirmed that ALV-J infection can induce mutation of p53 gene and increase mRNA expression [13,39], and some mutant forms of p53 might be involved in the development of tumors induced by ALV-J [40,41]. In the present study, p53 expression in ALV-J-SD1005-infected DF-1 cells was significantly increased by chicken 14-3-3σ over-expression and was significantly decreased by the knockdown expression. We speculate that the detected p53 might participate in the regulation of chicken 14-3-3σ on cell cycle in ALV-J infection as mutant forms. However, this speculation needs to be further examined.

## 5. Conclusions

In summary, it can be concluded that chicken 14-3-3σ, as a negative regulator, can cause G2/M-phase block and inhibit cell proliferation in ALV-J-infected cells by regulating the expression of CDK2/CDC2/p53; by contrast, ALV-J-SD1005 strain can promote cell proliferation by reducing chicken 14-3-3σ expressions, as shown in Figure 9. This study provides a new scientific dataset to elucidate the forming mechanism of ALV-J-induced fibrosarcoma.

## Figures and Tables

**Figure 1 viruses-15-00404-f001:**
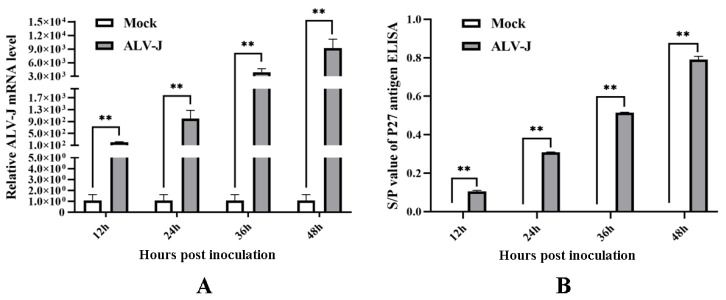
The proliferation viability of ALV-J-SD1005 strain in DF-1 cells at different times post-inoculation. (**A**) ALV-J gp85 mRNA expression levels in the cells were detected using qPCR; (**B**) ALV P27 antigen titers in the cells supernatant were detected using P27 antigen ELISA kit. **, *p* < 0.01.

**Figure 2 viruses-15-00404-f002:**
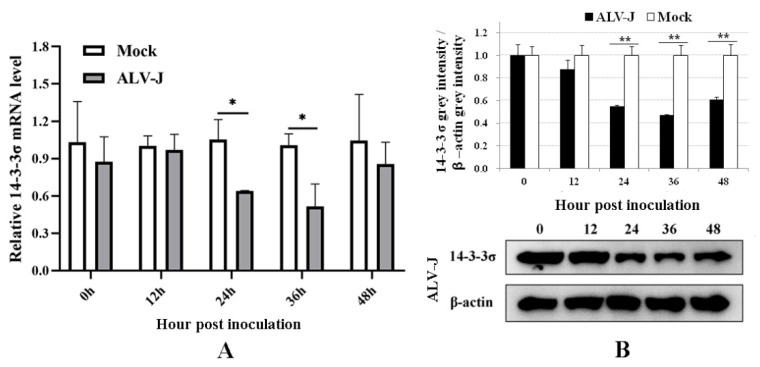
ALV-J-SD1005 strain inhibits the expressions of 14-3-3σ mRNA level (**A**) and protein level (**B**) in DF-1 cells. *, *p* < 0.05; **, *p* < 0.01.

**Figure 3 viruses-15-00404-f003:**
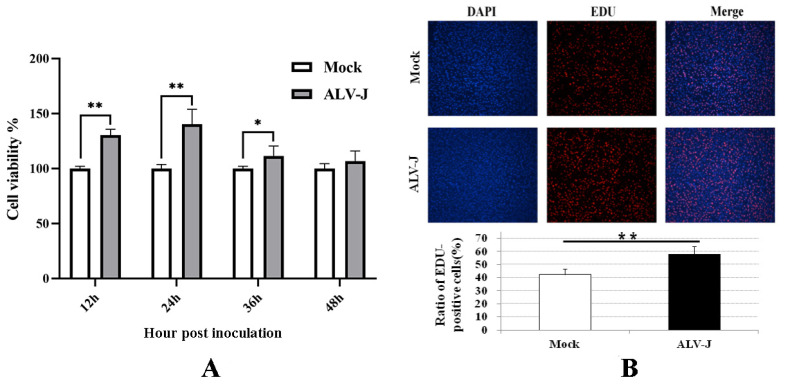
ALV-J-SD1005 strain promotes the proliferation of DF-1 cells. (**A**) The proliferation viability of DF-1 cells at different times post-virus inoculation was detected using CCK-8 method; (**B**) Proliferation viability of DF-1 cells at 36 hpi was detected using EDU fluorescence staining and microscopic examination. *, *p* < 0.05; **, *p* < 0.01.

**Figure 4 viruses-15-00404-f004:**
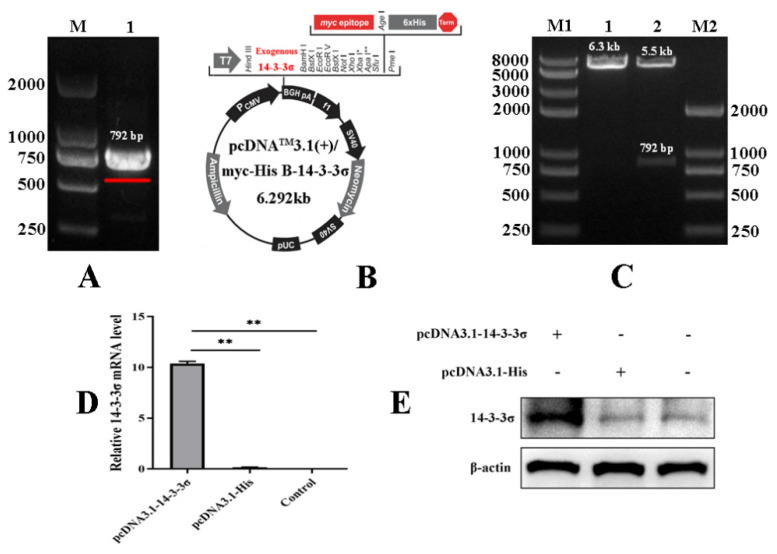
Construction of chicken 14-3-3σ over-expression plasmid and its bioactivity assay. (**A**) Amplification of chicken 14-3-3σ gene containing open reading frame; (**B**) schematic diagram of chicken 14-3-3σ over-expression plasmid; (**C**) identification of chicken 14-3-3σ over-expression plasmid by using both the enzyme (Hind III and BamH I) digestion tests; (**D**) mRNA expression level of chicken 14-3-3σ over-expression plasmid was detected using qPCR; (**E**) protein expression level of chicken 14-3-3σ over-expression plasmid was detected using Western blot. **, *p* < 0.01.

**Figure 5 viruses-15-00404-f005:**
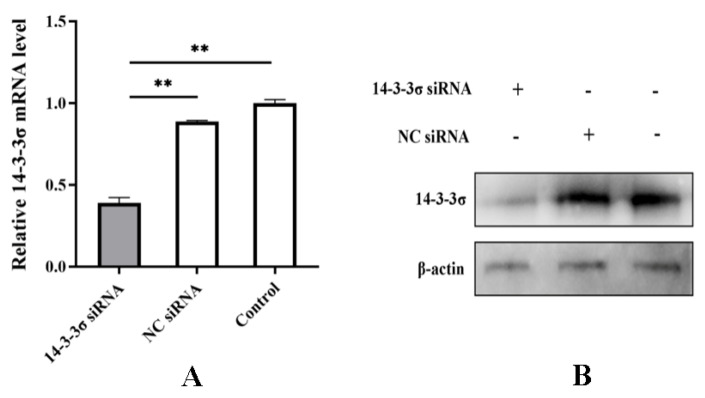
Bioactivity assay of chicken 14-3-3σ siRNA. (**A**) mRNA expression level of chicken 14-3-3σ was detected using qPCR; (**B**) protein expression level of chicken 14-3-3σ was detected using Western blot. **, *p* < 0.01.

**Figure 6 viruses-15-00404-f006:**
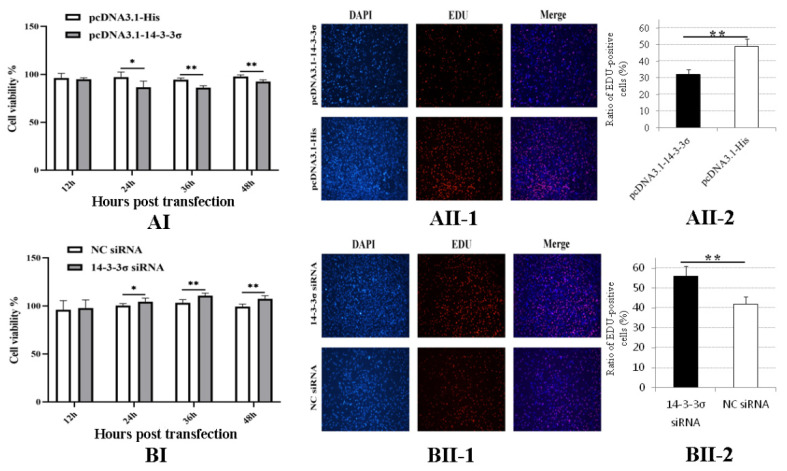
Effects of chicken 14-3-3σ over-expression and knockdown expression on the cell proliferation of ALV-J-infected DF-1 cells. (**AI**,**BI**), DF-1 cells proliferation viabilities at different hpt were detected using CCK-8; (**AII**,**BII**), DF-1 cells in division at 36 hpt were detected using EDU fluorescence staining and statistically analyzed. *, *p* < 0.05; **, *p* < 0.01.

**Figure 7 viruses-15-00404-f007:**
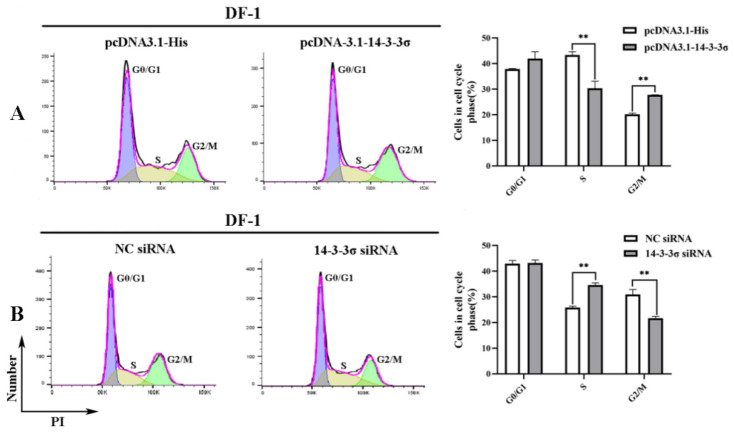
Regulation of chicken 14-3-3σ over-expression and knockdown expression on the division cycle of DF-1 cells infected with ALV-J-SD1005 strain. (**A** (**Left**)) and (**B** (**Left**)), DF-1 cell ratios of different division stages at 36 hpt were detected using flow cytometry; (**A** (**Right**)) and (**B** (**Right**)), DF-1 cell ratios of different cell cycles at 36 hpt were statistically counted. **, *p* < 0.01.

**Figure 8 viruses-15-00404-f008:**
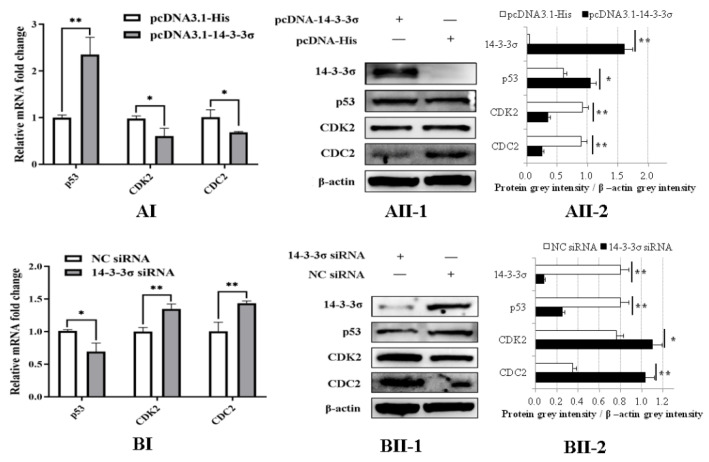
Regulations of chicken 14-3-3σ over-expression and knockdown expression on the expressions of p53/CDK2/CDC2 cyclin factors in ALV-J-infected DF-1 cells. (**AI**,**BI**), mRNA expressions of p53/CDK2/CDC2 cyclin factors at 36 hpt were detected using qPCR and statistically analyzed; (**AII**,**BII**), protein expressions of p53/CDK2/CDC2 cyclin factors at 36 hpt were detected using Western blot and statistically analyzed. *, *p* < 0.05; **, *p* < 0.01.

**Figure 9 viruses-15-00404-f009:**
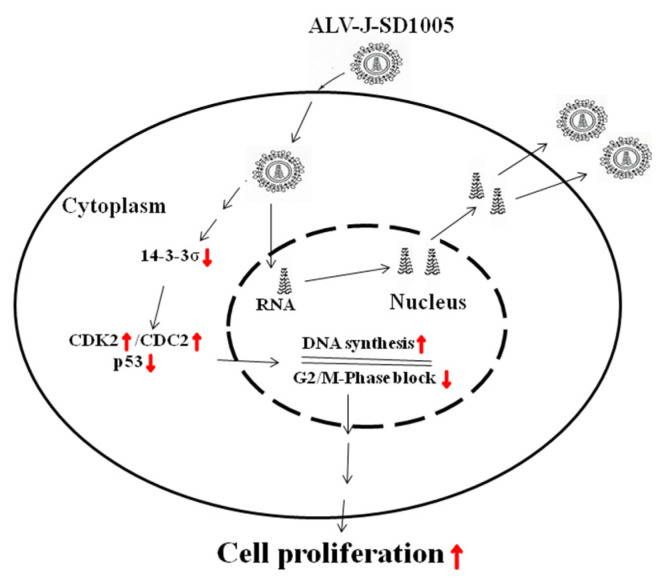
Pattern of ALV-J promoting cell proliferation via down-regulation of 14-3-3σ pathway. Red ↑, increase or promote; red ↓, decrease or inhibit.

**Table 1 viruses-15-00404-t001:** Sequences of the primers used for qPCR.

Primers	Sequences (5′-3′)	Primers	Sequences (5′-3′)
GAPDH-Fw	GCCATCACAGCCACACAGAAG	p53-Fw	GCACAGCCAAATCGGTCAC
GAPDH-Rv	GCAGGTCAGGTCAACAACAGA	p53-Rv	GCCACGTGCTCTGATTTCTTAT
14-3-3σ-Fw	GTGGTGCTGGGCTTGCT	CDK2-Fw	GTGACCTCAAACCCCAGAAC
14-3-3σ-Rv	CGTCTCCTTGCGGTCGTT	CDK2-Rv	TCCACAGCAGTCGAATAGTA
gp85-Fw	AACCAATCATGGACGATGGTA	CDC2-Fw	GAAAGTGAGGAGGAAGGTGT
gp85-Rv	TCCAAAGGTAAACCCATATGC	CDC2-Rv	CATGGAAAGGAATTCAAAAA

## Data Availability

The data that support the findings of this study are available from the corresponding author, [author initials], upon reasonable request.

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
