# Peer review of "J Subgroup Avian Leukosis Virus Strain Promotes Cell Proliferation by Negatively Regulating 14-3-3σ Expressions in Chicken Fibroblast Cells"

_viruses, 2023, doi:10.3390/v15020404_

Round 1

Reviewer 1 Report

In this study, Guo et.al have reported evidence of ALV-J promotes cell proliferation by reducing chicken 14-3-3σ expressions. This study is very interesting and important to understand the mechanism of tumorigenesis induced by ALV-J infection. However, there are some limitations.

1 Please reduce content of abstract.

2 Lines 312-313, the sentence “These results suggest that chicken 14-3-3σ can regulate the expressions of CDK2, CDC2 and P53 in ALV-J-infected DF-1 cells” was overstated and should be revised.

3 Section of discussion, please review the literature more deeply to compare your results with those of other authors..

Overall, the flow of this paper and grammar should be carefully checked when submission of revision version.

Author Response

A point-by-point Responses to Reviewer 1#

Manuscript ID: viruses-2164697

Title: J subgroup Avian Leukosis Virus Strain Promotes Cell Proliferation by Negatively Regulating 14-3-3σ Expressions in Chicken Fibroblast Cells

Journal: Viruses

We would like to thank the editor and the reviewers for your support and positive comments and suggestions given regarding the review of our manuscript. The comments are thoughtful and in-depth and definitely helped us to improve our manuscript. We have now revised the manuscript according to the reviewers' comments and suggestions in the revised manuscript with changes highlighted in yellow and provide a point-by-point response to the reviewers' comments. We believe that it is significantly improved. Please contact to me if you have any question.

Yours sincerely,

Reviewer 1#

In this study, Guo et.al have reported evidence of ALV-J promotes cell proliferation by reducing chicken 14-3-3σ expressions. This study is very interesting and important to understand the mechanism of tumorigenesis induced by ALV-J infection. However, there are some limitations. 

  1. Please reduce content of abstract.

Response: Thanks for your good correction. We have revised the abstract as following: This study focuses on clarifying the regulation of chicken 14-3-3σ protein on the fibrous histiocyte proliferation caused by ALV-J-SD1005 strain infection. DF-1 cells were inoculated with 102 TCID50 of ALV-J-SD1005 strain, and the cell proliferation viability was dramatically increased, and 14-3-3σ expressions were dramatically decreased within 48 hours post inoculation. Chicken 14-3-3σ over-expression could significantly decrease the cell proliferation and the ratio of S-phase cells, but increase the ratio of G2/M-phase cells in ALV-J-infected DF-1 cells; whereas, chicken 14-3-3σ knockdown-expression could cause the opposite effects. Additionally, chicken 14-3-3σ over-expression could also dramatically down-regulate the expressions of CDK2/CDC2, but up-regulate p53 expressions in the DF-1 cells; in contrast, the knockdown-expression could significantly increase the expressions of CDK2/CDC2 and decrease p53 expressions. It can be concluded that chicken 14-3-3σ can inhibit cell proliferation and cell cycle by regulating CDK2/CDC2/p53 expressions in ALV-J-infected DF1 cells; and ALV-J-SD1005 strain can promote cell proliferation by reducing 14-3-3σ expressions. This study helps for clarifying the forming mechanism of acute fibrosarcoma induced by ALV-J infection.

  1. Lines 312-313, the sentence “These results suggest that chicken 14-3-3σ can regulate the expressions of CDK2, CDC2 and P53 in ALV-J-infected DF-1 cells” was overstated and should be revised.

Response: Thanks for your good correction. We made changes in lines 314-315 as following: These results suggest that chicken 14-3-3σ can regulate mRNA level and protein level expressions of CDK2, CDC2 and p53 in ALV-J-infected DF-1 cells.

  1. Section of discussion, please review the literature more deeply to compare your results with those of other authors..

Response: Thanks for your good correction. We have revised the discussions and added some comparisons with those of other authors in the revised MS. The detail can be seen in the revised MS with highlighted yellow in the attachment.

Overall, the flow of this paper and grammar should be carefully checked when submission of revision version.

Response: We carefully checked the flow and grammar of this paper for many times and confirmed that it can be submitted.

Reviewer 2 Report

Subgroup J avian leukosis virus (ALV-J) can cause severe tumorigenesis and economic losses to the poultry industry. The ALV-J-SD1005 strain carrying v-fps oncogene can lead to acute the fibrosarcoma in vivo, but its tumorigenic process is not clear. In this study, the authors found that chicken 14-3-3σ protein could inhibit cell proliferation and cell cycle in  virus-infected DF-1 cells, while the ALV-J-SD1005 strain could promote cell proliferation by reducing the expression of chicken 14-3-3σ. The data in this manuscript is interesting, which contributes to the study of ALV-J-host interaction. However, there are still some points should be addressed in this version before it is accepted for publishment.

1. The abstract is a little too much, and some technical details of the experiment are not needed. Please modified the abstract.

2. Line 33, “Avian Leukemia virus” should be changed to “ avian leukosis virus.

3. In section 2.7 line 143,148,173; in section 3.5 line 344,345,253. The authors wrote "siRNA plasmids". siRNA is synthesized and transfected as RNA, not DNA plasmids, and plasmid-mediated knockdown is usually done by transfection of shRNA. please clarify the method used and correct it in the appropriate place.

4. In section 2.7 Table 1, the primers of beta-actin is absent while GAPDH’s seems to be redundant.

5. The statement of P53 in the text is not uniform; The expression of genes should be italicized, please check and correct them one by one.

6. In Figures 6, 8 and 9, "14-3-3δ" statement does not correspond to the cytokines described in this Ms, please correct them.

7. Some statements of difference significance sign “P” in the text and in the title of figures should be italicized, please check and correct them.

8. In Figure 4, the electrophoresis photograph of A is not clear, please improve or replace it.

9. In conclusions section, line 386-399, the font format of text is inconsistent with that of other parts, please correct it.

Author Response

A point-by-point response to Reviewer 2#

Manuscript ID: viruses-2164697

Title: J subgroup Avian Leukosis Virus Strain Promotes Cell Proliferation by Negatively Regulating 14-3-3σ Expressions in Chicken Fibroblast Cells

Journal: Viruses

We would like to thank the editor and the reviewers for your support and positive comments and suggestions given regarding the review of our manuscript. The comments are thoughtful and in-depth and definitely helped us to improve our manuscript. We have now revised the manuscript according to the reviewers' comments and suggestions in the revised manuscript with changes highlighted in yellow and provide a point-by-point response to the reviewers' comments. We believe that it is significantly improved. Please contact to me if you have any question.

Yours sincerely,

Reviewer 2

Subgroup J avian leukosis virus (ALV-J) can cause severe tumorigenesis and economic losses to the poultry industry. The ALV-J-SD1005 strain carrying v-fps oncogene can lead to acute the fibrosarcoma in vivo, but its tumorigenic process is not clear. In this study, the authors found that chicken 14-3-3σ protein could inhibit cell proliferation and cell cycle in  virus-infected DF-1 cells, while the ALV-J-SD1005 strain could promote cell proliferation by reducing the expression of chicken 14-3-3σ. The data in this manuscript is interesting, which contributes to the study of ALV-J-host interaction. However, there are still some points should be addressed in this version before it is accepted for publishment.

  1. The abstract is a little too much, and some technical details of the experiment are not needed. Please modified the abstract.

Response: Thanks for your good suggestions. We have revised the abstract as following: This study focuses on clarifying the regulation of chicken 14-3-3σ protein on the fibrous histiocyte proliferation caused by ALV-J-SD1005 strain infection. DF-1 cells were inoculated with 102 TCID50 of ALV-J-SD1005 strain, and the cell proliferation viability was dramatically increased, and 14-3-3σ expressions were dramatically decreased within 48 hours post inoculation. Chicken 14-3-3σ over-expression could significantly decrease the cell proliferation and the ratio of S-phase cells, but increase the ratio of G2/M-phase cells in ALV-J-infected DF-1 cells; whereas, chicken 14-3-3σ knockdown-expression could cause the opposite effects. Additionally, chicken 14-3-3σ over-expression could also dramatically down-regulate the expressions of CDK2/CDC2, but up-regulate p53 expressions in the DF-1 cells; in contrast, the knockdown-expression could significantly increase the expressions of CDK2/CDC2 and decrease p53 expressions. It can be concluded that chicken 14-3-3σ can inhibit cell proliferation and cell cycle by regulating CDK2/CDC2/p53 expressions in ALV-J-infected DF1 cells; and ALV-J-SD1005 strain can promote cell proliferation by reducing 14-3-3σ expressions. This study helps for clarifying the forming mechanism of acute fibrosarcoma induced by ALV-J infection.

  1. Line 33, “Avian Leukemia virus” should be changed to “ avian leukosis virus”.

Response: Thanks for your good correction. We have revised it in Line 29 and Line 298 as following : Avian leukosis virus (ALV) is a tumorigenic retrovirus and is divided into 11 subgroups named from A to K [1,2].  

ALV-J-SD1005 strain from the fibrosarcoma tissue of "817" broiler breeder chickens in Shandong belongs to subgroup J avian leukosis virus [26].

  1. In section 2.7 line 143,148,173; in section 3.5 line 344,345,253. The authors wrote "siRNA plasmids". siRNA is synthesized and transfected as RNA, not DNA plasmids, and plasmid-mediated knockdown is usually done by transfection of shRNA. please clarify the method used and correct it in the appropriate place.

Response: Thank you for your good questions. We used a synthesized small interfering RNA (siRNA) to knockdown the expression of chicken 14-3-3σ in this study and corrected the statement on that in the revised MS as following:

Line 129: A small interfering RNA (siRNA) against chicken 14-3-3σ (14-3-3σ siRNA) with the target sequence:

Line 132: pcDNA3.1-14-3-3σ plasmids, 14-3-3σ siRNA and negative control siRNA (5'-UUCUCCGAACGUGUCACGUTT-3') were transfected

Line 157: The virus-infected DF-1 cells in 6-well plates were transfected with pcDNA3.1-14-3-3σ plasmid, pcDNA3.1-His plasmid, 14-3-3σ siRNA and NC siRNA, respectively;

Line 227: 3.5. Preparation of the siRNA against chicken 14-3-3σ

Line 228:An interfering RNA (siRNA) against the 161st base site of chicken 14-3-3σ gene (14-3-3σ siRNA) and……

Line 236: Figure 5 Bioactivity assay of chicken 14-3-3σ-siRNA

Line 244: and after 12 hpi, pcDNA3.1-14-3-3σ plasmid and 14-3-3σ siRNA were transfected into the DF-1 cells, respectively;

Line 315-316: Then, a small interfering RNA used for knockdown-expression of chicken 14-3-3σ was designed and synthesized.

  1. In section 2.7 Table 1, the primers of beta-actin is absent while GAPDH’s seems to be redundant.

Response: Thanks for your kind reminding. We use the GAPDH gene as an internal reference gene in qPCR, so its primers in this study is necessary; but beta-actin is used as an internal reference protein in Western blot, so its primers is not necessary to list in the Table.

  1. The statement of p53 in the text is not uniform; The expression of genes should be italicized, please check and correct them one by one.

Response: Thanks for your kind correction. We have corrected them one by one in the revised MS. The file can be seen in the attachment.

  1. In Figures 6, 8 and 9, "14-3-3δ" statement does not correspond to the cytokines described in this Ms, please correct them.

Response: Thanks for your kind correction. We have corrected them one by one in Figures 6, 8 and 9 in the revised MS. The file can be seen in the attachment.

  1. Some statements of difference significance sign “P” in the text and in the title of figures should be italicized, please check and correct them.

Response: Thanks for your kind correction. We have have corrected them one by one in the revised MS. The filel can be seen in the attachment.

  1. In Figure 4, the electrophoresis photograph of A is not clear, please improve or replace it.

Response: Thanks for your good question. We replaced it with better electrophoresis photograph in Figure 4. The file can be seen in the attachment.

  1. In conclusions section, line 386-399, the font format of text is inconsistent with that of other parts, please correct it.

Response: Thanks for your good correction. We have revised them in the revised MS. The file can be seen in the attachment.

                          Authors: Wang &Guo et al.